# Biochar with Inorganic Nitrogen Fertilizer Reduces Direct Greenhouse Gas Emission Flux from Soil

**DOI:** 10.3390/plants12051002

**Published:** 2023-02-22

**Authors:** Muhammad Ayaz, Dalia Feizienė, Vita Tilvikienė, Virginijus Feiza, Edita Baltrėnaitė-Gedienė, Sana Ullah

**Affiliations:** 1Institute of Agriculture, Lithuanian Research Centre for Agriculture and Forestry, Instituto al. 1, Akademija, LT-58344 Kėdainiai, Lithuania; 2Institute of Environmental Protection, Vilnius Gediminas Technical University, LT-10223 Vilnius, Lithuania

**Keywords:** biochar, CO_2_, N_2_O, CH_4_ emissions, cumulative emissions, global warming potential, soil moisture, soil temperature

## Abstract

Agricultural waste can have a catastrophic impact on climate change, as it contributes significantly to greenhouse gas (GHG) emissions if not managed sustainably. Swine-digestate-manure-derived biochar may be one sustainable way to manage waste and tackle GHG emissions in temperate climatic conditions. The purpose of this study was to ascertain how such biochar could be used to reduce soil GHG emissions. Spring barley (*Hordeum vulgare* L.) and pea crops in 2020 and 2021, respectively, were treated with 25 t ha^−1^ of swine-digestate-manure-derived biochar (B_1_) and 120 kg ha^−1^ (N_1_) and 160 kg ha^−1^ (N_2_) of synthetic nitrogen fertilizer (ammonium nitrate). Biochar with or without nitrogen fertilizer substantially lowered GHG emissions compared to the control treatment (without any treatment) or treatments without biochar application. Carbon dioxide (CO_2_), nitrous oxide (N_2_O), and methane (CH_4_) emissions were directly measured using static chamber technology. Cumulative emissions and global warming potential (GWP) followed the same trend and were significantly lowered in biochar-treated soils. The influences of soil and environmental parameters on GHG emissions were, therefore, investigated. A positive correlation was found between both moisture and temperature and GHG emissions. Thus, biochar made from swine digestate manure may be an effective organic amendment to reduce GHG emissions and address climate change challenges.

## 1. Introduction

In recent decades, the increase in human population has caused serious challenges to the agriculture sector and to the agronomist in ensuring food security, causing minimum soil and environmental pollution [1]. Inorganic nitrogen fertilizer consumption in the agricultural sector of the European Union has increased by around 2% over the past ten years to 10.2 million tons [2]. This considerable share of synthesized fertilizer application is due to inefficient use, which causes financial harm, environmental damage, and health risks [3,4,5]. Inorganic fertilizer amendment and soil tillage practices have increased greenhouse gas (GHG) emissions [6,7,8]. By 2030, it is anticipated that the agricultural sector’s nitric oxide (N_2_O) emissions may rise by 35–60%. This increase is linked to higher nitrogen content due to fertilizer use and higher production of animal waste [9,10]. Moreover, the increase in the number of livestock is directly proportional to methane (CH_4_) emissions that, between 1990 and 2030, are anticipated to increase by 60% [11]. The predicted rise in agrofarming emissions is 8–8.4%, with a mean increase of 8.3 Pg CO_2_-eq by 2030, assuming the aforesaid rates of rising emissions (10–15%) for the 2020–2030 period. Anthropogenic emissions of GHGs (CO_2_, CH_4_, and N_2_O), have become a substantial contributor to global climate change [12,13]. GHG emissions are highly dependent on soil temperature and moisture, which are instantaneously affected by biochar application [14,15,16]. The use of biochar is a farming method that improves soil temperature and moisture retention. There are two ways to best manage increase in soil moisture: (1) to increase the amount of organic matter in the soil (because the stability and porosity of aggregates are improved by the direct addition of biochar [17,18]) and (2) to increase physical barriers on the surface, which reduces soil surface runoff and evaporation [19]. Thus, soil moisture and temperature improvements appear to balance the emission of GHGs. Biochar application enhances soil organic carbon and, consequently, results in carbon sequestration [20,21]. Biochar amendment boosts crop productivity and reduces the overuse of synthetic chemical fertilizers while simultaneously increasing soil organic carbon (SOC) stock, moisture retention, and temperature [22,23,24]. By reducing the usage of synthetic fertilizers, this strategy can enhance both environmental and human health [25,26].

One of the main economic drivers is agriculture, and according to EEA Report No. 17, the sector’s overall GHG emissions, which amount a 4.4 Mt CO_2_ equivalent, are slightly lower than those of the transportation sector [27]. According to the long-term commitment EU panel, which was initiated in 2014 by Fertilizers Europe, the EU mineral fertilizer industry promotes fertilizer use efficiency to improve agricultural production and a sustainable environment (EU Nitrogen Expert Panel). Moreover, the addition of biochar improves the efficiency of the bacteria involved in carbon and nitrogen metabolism, considerably enhancing CO_2_, CH_4_, and N_2_O outputs [28,29]. However, using biochar in conjunction with inorganic nitrogen fertilizer lowers GHG emissions without reducing crop output [30,31]. The impact of adding swine-digestate-derived biochar as an organic amendment with nitrogen fertilizer on GHG emissions from the soil in spring barley and pea cropping systems in the EU has not yet been reported. The use of biochar made from swine digestate is, therefore, hypothesized to minimize GHG emissions, either with or without using 70% of the recommended dose of synthetic nitrogen fertilizer. Therefore, two-year experiments on the mitigation of GHGs coupled with reduced synthetic nitrogen fertilizer are carried out at the research fields of the Lithuanian Research Center for Agriculture and Forestry, Lithuania. This study’s primary goals are to decrease the excessive use of inorganic synthetic nitrogen fertilizer, improve environmental quality by lowering GHG emissions, and evaluate the usage of biochar made from swine digestate as a potential substitute source of fertilizer.

## 2. Materials and Methods

### 2.1. Experimental Site

The experimental study was conducted during the growing seasons of 2020 and 2021 at the fields of the Lithuania Research Center for Agriculture and Forestry (55°40′ N, 23°87′ E). The chemical compositions at depths of 0–10, 0–20, and 0–60 cm for the Endocalcari-Epihypogleyic Cambisol soil used in the experimental fields is shown in Table 1. Biochar was prepared from swine manure digestate at 550 °C. Both biochar and N fertilizer were applied to the soil one week prior to sowing of the crop and were manually applied to each plot (1.5 m^2^). The experiment was carried out using the spring barley (*Hordeum vulgare* L.) “Luoke cultivar” and the pea (*Pisum sativum*) “Respect cultivar” in 2020 and 2021, respectively. The period of growth was from April to August 2020 for spring barley and from April to July 2021 for the pea crop. Data were recorded at each growth stage from seedling until maturity. Lithuanian Hydrometeorological Service-Dotnuva data under the Ministry of Environment data were used (http://www.meteo.lt/, accessed on 12 January 2020) (Figure 1). The chemical changes over two years in the fields studied are given in Table 1.

### 2.2. Soil Physicochemical Properties

Laboratory-based, standardized techniques were used to examine the physicochemical characteristics of both the soil and the biochar. A 1:5 (vol:vol^−1^) soil combination in 1 M KCl solution was used for the electrical conductivity and pH analysis of the soil and biochar [32,33], as well as an extract in distilled water [34]. Soil and biochar organic matter contents were measured using a spectrophotometer at a wave length of 590 nm [35]. A revised ammonium-acetate technique was used to measure cation exchange capacity [36]. Inductively coupled plasma atomic emission spectrometry (Perkin Elmer ICP-OES, Waltham, MA, USA) was used to assess the extractable Mg from DTPA [37]. Using a reference approach, total nitrogen (TN) and accessible phosphorus concentrations were determined [38]. A TGA provided information on biochar ash content, moisture, volatiles, and residual mass.

### 2.3. Experimental Design

The field study used a three-factorial randomized full-block design with six treatments and three replications. No biochar plus no nitrogen fertilizer (control), no biochar plus 160 kg ha^−1^ nitrogen fertilizer (N_1_), no biochar plus 120 kg ha^−1^ nitrogen fertilizer (N_2_), 25 t ha^−1^ biochar plus 160 kg ha^−1^ nitrogen fertilizer (N_1_B), 25 t ha^−1^ biochar plus 120 kg ha^−1^ nitrogen fertilizer (N_2_B), and 25 t ha^−1^ biochar plus no nitrogen fertilizer (N_2_B) were the experimental treatments (B).

### 2.4. Gas Sampling and Flux Calculation

Gas chromatography was used to measure the gas flux, and a static chamber gas [13,39] method was modified only slightly for the analysis. A U-shaped groove (50 mm wide and 50 mm deep) was present on the top edge of the chamber base box (frame) to retain a detachable chamber box. Stainless-steel frames were permanently buried 10 cm beneath the surface of the soil. A frame’s perimeter covered 0.36 m^2^. The chamber was sealed for 3 min before each flux measurement, and 20 mL of gas sample was drawn using a 20 cm^3^ syringe. To increase the consistency of gaseous flux estimates, the gas samples were collected between the hours of 9:00 and 10:00 in the morning. Glass vials with rubber tubing used as a lid were used to collect the gas samples. From the beginning of the cultivating season (one week before the application of biochar) to one month after harvest, the fluxes of CO_2_, N_2_O, and CH_4_ were measured at 2-week intervals. Three replicates of each treatment were used; thus, gas samples were collected from each plot. The samples were examined using a gas chromatograph (HP 6890 Series, GC System, Hewlett-Packard, Analytical system Management, Denver, USA) that had nickel catalysts for converting CO_2_ to CH_4_ and flame ionization and electron capture detectors. The corresponding temperatures were 70, 300, and 350 °C, respectively. The techniques for gas chromatography were explained by [40]. Equations (1) and (2) were used, respectively, to compute the cumulative and GHG flow rates and global warming potential for the growing seasons of 2020 and 2021 (from April to August). Based on the rate of change in GHG concentration within the chamber, which was determined as the slope of the linear regression between the GHG concentration and the gas-sampling time, the flow rate of each GHG was derived.
(1)$$R_{a}=\left[\frac{R_{i}+R_{i}+1}{2}\right]\times n\;$$
(2)$$\mathrm{GWP}\;=\left({\mathrm{CO}}_{2}*1\right)+\left(N_{2}O*\;298\right)+\left({\mathrm{CH}}_{4}*25\right)$$

### 2.5. Calculation of Cumulative Soil GHG Emissions

Between various growth stages, cumulative CO_2_, CH_4_, and N_2_O emissions for each treatment were estimated as indicated by [5,41].

The total cumulative emissions of soil CO_2_, CH_4_, and N_2_O (mgha^−1^h^−1^) are represented by the symbol R_a_, where the initial emissions of soil CO_2_, CH_4_, and N_2_O are represented by R_i_; the subsequent emissions of soil CO_2_, CH_4_, and N_2_O are represented by R_i_ + 1 after the subsequent time i; and n is the number of interval days for the emissions of soil CO_2_, CH_4_, and N_2_O.

### 2.6. Global Warming Potential (GWP)

The following equation was used to determine the global warming potential (GWP) of soils treated with biochar and N fertilizer in 2020–2021 (IPCC, 2007).

### 2.7. Soil Temperature and Moisture Measurements

In each growth stage, the soil temperature was measured using a squarely buried thermometer at a depth of 5 cm over the years 2020–21 [5]. Additionally, using an oven-drying method for 24 h at 105 °C, soil samples were taken from 0 to 10 cm deep using a soil auger in order to quantify soil moisture (in mass percent) at each growth stage. The link between soil temperature, moisture content, and CO_2_, N_2_O, and CH_4_ emissions was examined using a linear regression.

### 2.8. Statistical Analysis

An analysis of variance (ANOVA) was performed on annual data gathered for each parameter during the 2-year period. The statistical differences were examined using Statistix 8. The Tukey Test was used to assess mean values at a *0.05* probability level. GraphPad Prism 9 was used to plot the data.

## 3. Results

### 3.1. Soil CO_2_ Emission Flux

Periodic variation was recorded between the seasons of the spring barley crop of 2020 and the pea crop of 2021. The emission of CO_2_ during the 2020 spring barley crop was recorded as higher throughout the season, except during jointing stage. During 2021 of the pea crop season, CO_2_ emissions were found significantly (≤0.05) lowered under biochar-treated soils compares to the spring barley of 2020 (Figure 2). All biochar-treated soils showed substantially (≤0.05) lowered CO_2_ emissions (by 57%, 55%, and 59%, respectively, for B, N_1_B, and N_2_B) compared to non-biochar-treated soils during all stages of the pea crop. The N_2_B treatment significantly (≤0.05) lowered CO_2_ emissions during the tillering stage (by 58%) compared to the control treatment. Similarly, the B treatment substantially (≤0.05) lowered CO_2_ emissions during the jointing, flowering, and maturity stages by 50%, 51%, and 50%, respectively (Figure 2).

### 3.2. Soil N_2_O Emission Flux

Biochar treatments had no effect on N_2_O emissions during the spring barley crop of 2020 throughout all the growth stages. However, there was significant variation recorded during different growth stages of both cropping seasons regarding N_2_O emissions (Figure 3). Following the above trend, the biochar-treated soils of B, N_1_B, and N_2_B showed substantially (≤0.05) lowered N_2_O emissions by 48%, 49%, and 48%, respectively, throughout the growth stages of the pea crop compared to non-biochar treatments (Figure 3). In 2021, during the pea crop season, the N_2_ treatment (100% nitrogen fertilizer alone) tended to enhance N_2_O emission, specifically during the seedling and tillering stages.

### 3.3. Soil CH_4_ Emission Flux

There was no substantial variation recorded for methane gas (CH_4_) emissions during the spring barley crop of 2020. However, the results indicated significant fluctuation in CH_4_ emissions during the pea crop growing stages of 2021 (Figure 4). The N_1_B treatment significantly (≤0.05) lowered CH_4_ emissions by 17%, and 19% during the flowering and maturity stages, respectively (Figure 4). However, biochar did not affect CH_4_ emissions the way it effected CO_2_ and N_2_O emissions.

Biochar application did not affect the cumulative emissions of CO_2_, N_2_O, and CH_4_ during the growth stages of the spring barley crop of 2020 (Table 1). However, there was big variation recorded during the growth stages of the pea crop of 2021. The biochar treatments of B, N_1_B, and N_2_B significantly (≤0.05) lowered cumulative CO_2_ emissions by 16%, 19%, and 17%, respectively, compared to the control treatment. However, the cumulative emission of N_2_O was significantly (≤0.05) lowered by 6% only in the B treatment compared to the control treatment (Table 2). Similarly, the cumulative emission of CH_4_ was significantly lowered by 7% in the B and N_1_B treatments compared to the control treatment (Table 2).

The global warming potential (GWP) of CO_2_, N_2_O, and CH_4_ emissions followed the same trend as that of cumulative emissions. There was no significant fluctuation recorded for GWP during the growth stages of spring barley in 2020. However, the GWP caused by CO_2_ was recorded as substantially (≤0.05) lower in the biochar treatments of B, N_1_B, and N_2_B by 39%, 35%, and 39%, respectively, compared to other treatments during the growth stages of the pea crop of 2021 (Table 3). The GWPs caused by CH_4_ and N_2_O were significantly lowered in treatment B by 19% and 34% compared to the control treatment, respectively (Table 3).

The substantial changes under biochar and N fertilizer rates for soil moisture and temperature are presented in (Figure 5). During 2020–2021, no significant effects of the treatments were recorded for lower soil temperature during the vegetative growth stages of both crops. Furthermore, soil moisture contents were 9.5%, 8.3%, and 7.6% higher in the N_2_B, N_1_B, and B treatments, respectively, during flowering and maturity stages of spring barley. A similar trend was recorded during different growth stages of the pea crop (Figure 5).

## 4. Discussion

### 4.1. CO_2_ Emission

The decomposition of organic materials is caused by CO_2_, N_2_O, and CH_4_ emissions [42,43]. Biochar provides additional environmental advantages since it improves soil fertility through decomposition [44,45]. Biochar is a significant source of carbon and helps in increasing SOC buildup [46,47] that, even at low soil temperatures, resulted in higher average CO_2_ emissions in biochar-amended soil compared to non-biochar treatments in 2020. This result showed that soil C has a greater potential for soil CO_2_ emission variability [48,49] and, hence, increases soil fertility. However, on the other hand, biochar has the potential to mitigate CO_2_ emission [50,51]. The seasonal changes in soil CO_2_ were dramatically impacted by biochar applications in 2021. Similar fluctuations in soil temperature and moisture were visible for the biochar treatments during the field trial. The agricultural fields′ consistent results demonstrate that biochar decomposition initially boosted soil CO_2_ emission and soil carbon and nitrogen availability [52,53]. However, there was also substantially lowered soil CO_2_ after biochar decomposition compared to non-biochar treatments. Unnecessary agronomic practices may influence soil moisture, which can affect soil CO_2_ emission [54,55]. For instance, it was reported that different tillage operations lead to GHG emissions [56]. According to the current study, the higher precipitation (Figure 1) in the first year (2020) compared with that of the second year (2021) could mean that, due to favorable conditions for the decomposition of biochar, the soil CO_2_ emission increased [57,58].

### 4.2. N_2_O Emission

The overusage of N fertilizer increases GHG emissions and has negative effects on the ecosystem [59,60]. The current findings showed that, during the growth seasons, the N fertilizer treatments of N_1_, N_2_, and N_3_ alone considerably boosted soil N_2_O emissions. (2021). Based on the N used and the emission variables, soil amendment with biochar and synthetic fertilizers can reduce N_2_O emissions [61]. It is challenging to anticipate the emission factors because of the complex chemical compositions of organic fertilizers [5,62]. It is known that N fertilization and mulching treatments together boost N_2_O flux by 71–123% [63,64,65].

Nevertheless, biochar on a field might help to reduce N_2_O emissions [66,67]. Additionally, it was determined from these outcomes that N application with biochar could decrease N_2_O emissions, as observed in the biochar-treated plots compares to non-biochar-treated plots. Moreover, it was also reported that N fertilization could influence degradable N and C, which resulted in improving the intricate microbial interaction between N and C, thus enhancing N_2_O emissions [68,69].

As a comparison to applying N fertilizer alone, using biochar with N fertilizer reduces N_2_O emissions by 25–35% [70,71]. Higher nitrogen fertilizer application rates result in higher GHG emissions, which has an immediate impact on soil N_2_O emissions [5,72,73]. The present study suggested that the N_1_, N_2_, and N_3_ treatments are more environmentally unfriendly due to N_2_O emissions, while the B treatment with N fertilizer is ecofriendly.

### 4.3. CH_4_ Emission

Compared to CO_2_ and N_2_O emissions, only the N_1_B treatment decreased CH_4_ emissions during 2021. It was reported that the organic piles’ structure was improved with biochar application anaerobically, and biochar could alter the oxidation–reduction potential by enhancing absorptivity, which lowered the mechanism of methanogens and increased that of methanotrophs to mitigate CH_4_ emissions [74]. Several of the literature findings have indicated that the interaction between applying biochar to soil and CH_4_ flux is not well-known [75,76]. The soil applications of biochar have been shown to enhance [77], lower [77,78,79], or have no substantial influence on CH_4_ emission flux [80]. It was reported that biochar addition to soil also promoted methanotrophic CH_4_ intake at the oxic–anoxic junction in anaerobic environments. Moreover, the addition of biochar improved the oxidation of CH_4_ by methanotrophic organisms at the oxic–anoxic root interface, which lowered the concentration of CH_4_ that could enter a plant′s aerenchyma and escape [79].

### 4.4. Global Warming Potential (GWP)

The overall impact of the main greenhouse gases (i.e., CO_2_, CH_4_, and N_2_O) is driven by GWP [81]. The plots with applied biochar had a much lower net GWP during 2021. However, no substantial difference was reported in 2020, which is in line with the following reports. The non-significant difference in GWP might be due to the non-decomposition of biochar in the first year [82]; however, the decomposition of biochar might be enhanced in the second year, which led to GWP reduction [51]. Overall, studies report that biochar application can significantly mitigate global warming. The biochar C:N ratio may be an important factor that drives GWP under biochar applications [83].

### 4.5. Soil Moisture and Temperature

The incorporation of biochar into soil is treated as sustainable waste. It was reported that the inappropriate management of different wastes (food, agriculture, etc.) creates a global environmental challenge [84]. Thus, biochar addition provides a multitude of advantages in terms of sustainable environment and agriculture aspects [85]. Furthermore, the current study reported that biochar applications significantly (*p* ≤ 0.05) elevated soil moisture content in the years of 2020–2021. Greater soil moisture content as a result of surface area and porous structure has been observed [22,86]. However, changes in soil temperature could be attributed to weather conditions. 

### 4.6. Correlation between Soil Moisture, Temperature, and CO_2_ and N_2_O Emissions

According to the current study, CO_2_, N_2_O, and CH_4_ emissions were considerably positively associated with soil temperature and moisture. According to a report, the primary variables affecting soil gas emission fluxes are its thermal characteristics [87,88]. Soil CO_2_ and CH_4_ emissions increase due to fact that higher soil temperature and moisture cause higher biochar decomposition and higher methane oxidation rates [89,90,91]. The reason for higher N_2_O emissions with higher temperatures could be attributed to N fertilization, which releases mineralized N upon decomposition [92]. It was observed that soil temperature and soil moisture had a positive, two-parameter linear association with CO_2_, N_2_O, and CH_4_ emissions (Figure 6). The findings of this study, therefore, show that agricultural management techniques under humid climate conditions affected the rate of soil GHG emission. 

For the spring barley and pea crops in 2020–2021, the linear relationship between soil CO_2_, N_2_O, and CH_4_ emissions and soil moisture and soil temperature was studied. Figure 6 demonstrates a positive correlation between soil temperature and moisture and soil GHG CO_2_, N_2_O, and CH_4_ emissions. The linear CO_2_, N_2_O, and CH_4_ R^2^ values during 2020 and 2021 were 0.2171 and 0.1353, 0.5550 and 0.3355, and 0.7611 and 0.1981, respectively. According to Figure 6, soil CO_2_, N_2_O, and CH_4_ emissions significantly increased when soil temperature and wetness rose.

## 5. Conclusions

Biochar application substantially lowered direct CO_2_, N_2_O, and CH_4_ emissions from soil in the second year compared to first year for non-biochar-treated plots. Thus, the lower CO_2_, N_2_O, and CH_4_ emissions from the agricultural fields confirmed that swine manure digestate biochar could be a suitable remedy for agriculture fields with higher GHG emissions, especially in temperate climatic conditions. Likewise, the cumulative emissions and global warming potential were substantially influenced by biochar during the second year of the experiment. A positive correlation was recorded between GHG emissions and soil moisture and temperature. No negative environmental issues were recorded during the two years of field research. More research is required to explore the long-term implication of swine-digestate-manure-derived biochar.

## Figures and Tables

**Figure 1 plants-12-01002-f001:**
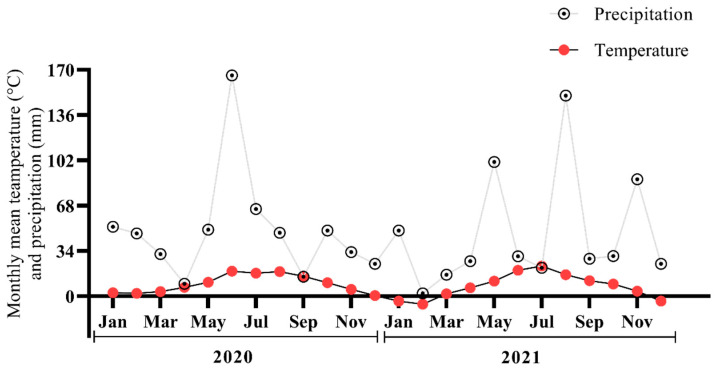
Mean monthly precipitation (mm) and air temperature (°C) during 2020 and 2021.

**Figure 2 plants-12-01002-f002:**
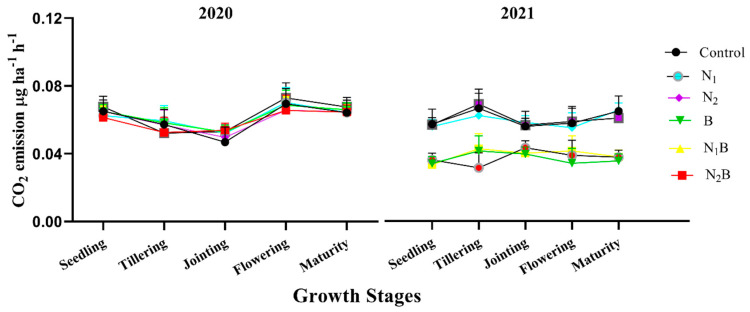
Direct effect of biochar on CO_2_ emissions at different growth stages of spring barley (2020) and after effects on pea crop (2021). Treatments: control-without amendments; N_1_-160 kg ha^−1^ nitrogen fertilizer; N_2_-120 kg ha^−1^ nitrogen fertilizer; B-25 t ha^−1^ biochar; N_1_B-160 kg ha^−1^ nitrogen fertilizer plus 25 t ha^−1^ biochar; N_2_B-120 kg ha^−1^ nitrogen fertilizer plus 25 t ha^−1^ biochar.

**Figure 3 plants-12-01002-f003:**
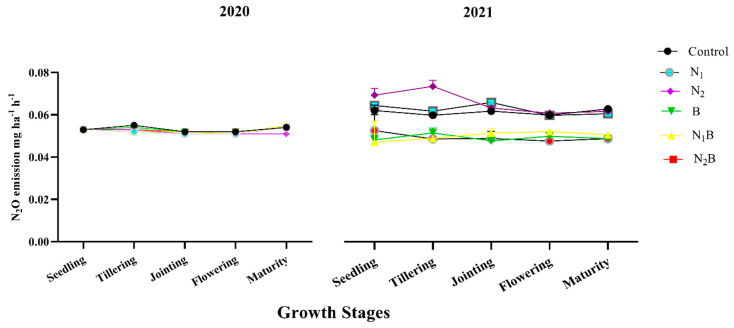
Direct effect of biochar on N_2_O emissions at different growth stages of spring barley (2020) and after effects on pea crop (2021). Treatments: control-without amendments; N_1_-160 kg ha^−1^ nitrogen fertilizer; N_2_-120 kg ha^−1^ nitrogen fertilizer; B-25 t ha^−1^ biochar; N_1_B-160 kg ha^−1^ nitrogen fertilizer plus 25 t ha^−1^ biochar; N_2_B-120 kg ha^−1^ nitrogen fertilizer plus 25 t ha^−1^ biochar.

**Figure 4 plants-12-01002-f004:**
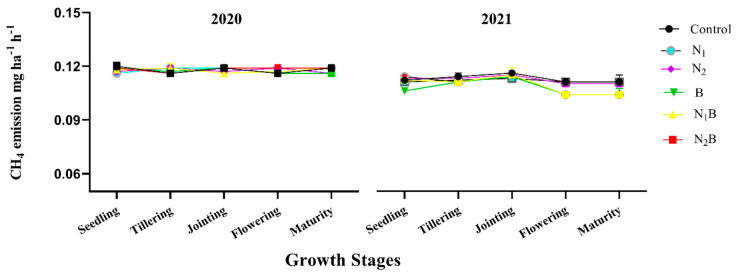
Direct effect of biochar on CH_4_ emissions at different growth stages of spring barley (2020) and after effects on pea crop (2021). Treatments: control-without amendments; N_1_-160 kg ha^−1^ nitrogen fertilizer; N_2_-120 kg ha^−1^ nitrogen fertilizer; B-25 t ha^−1^ biochar; N_1_B-160 kg ha^−1^ nitrogen fertilizer plus 25 t ha^−1^ biochar; N_2_B-120 kg ha^−1^ nitrogen fertilizer plus 25 t ha^−1^ biochar.

**Figure 5 plants-12-01002-f005:**
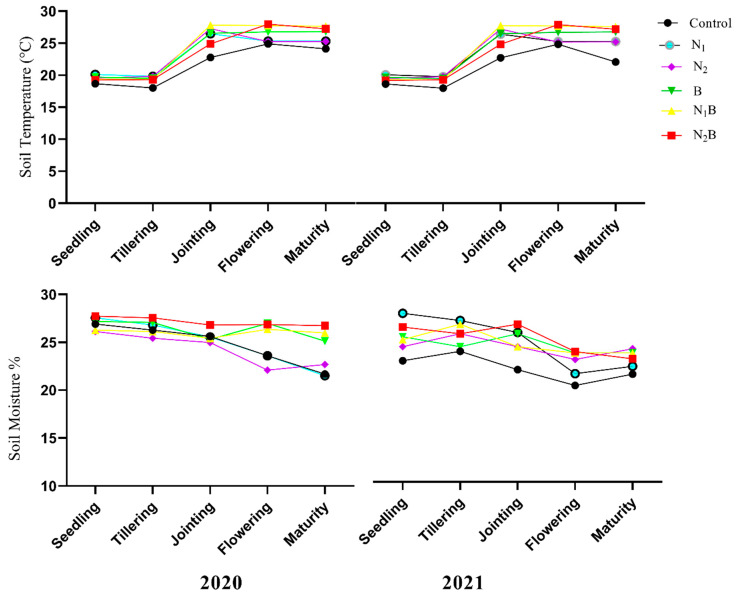
Changes in soil moisture content (%) and soil temperature (°C) in different growth stages of spring barley (2020) and pea crops (2021).

**Figure 6 plants-12-01002-f006:**
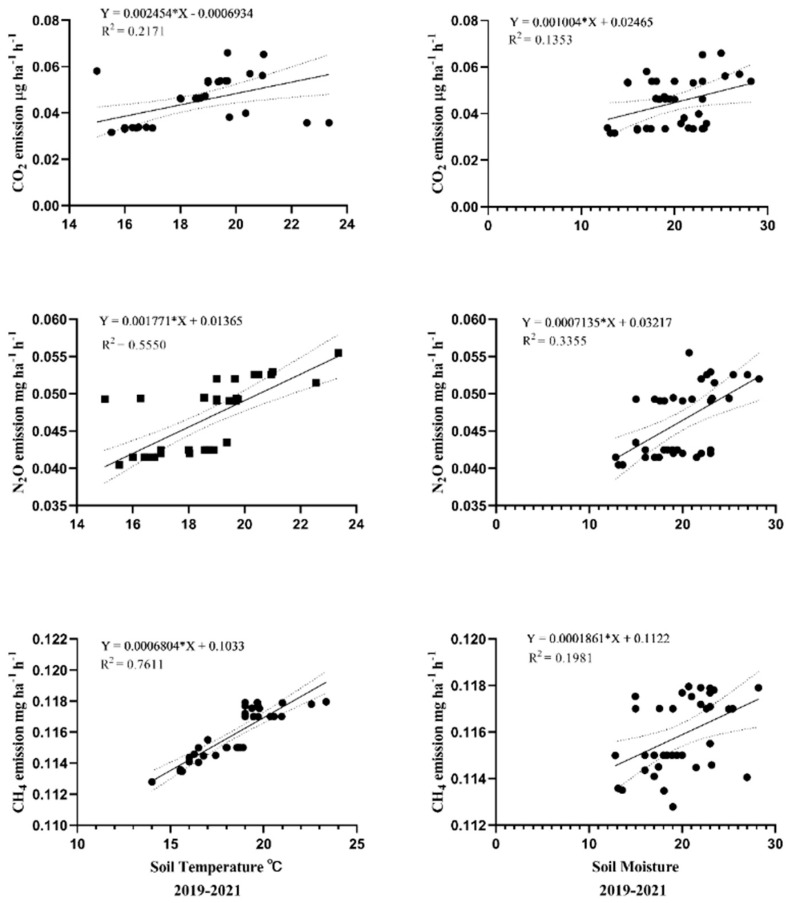
Linear relationship of soil CO_2_, N_2_O, and CH_4_ emissions with soil temperature and soil moisture at different growth stages of spring barley (2020) and pea crops (2021).

**Table 1 plants-12-01002-t001:** Physiochemical properties of soil and biochar and soil chemical changes during two years.

Soil								
Depth (cm)	pH	Total N (%)	P_2_O_5_ (mg/kg)	K_2_O (mg/kg)	Organic carbon (%)	Mineral nitrogen (mg/kg)	NH_4_-N (mg/kg)	NO_3_-N+N_2_O-N (mg/kg)
0–10	6.8	0.14	13.90	228.22	1.02	-	-	-
0–20	6.9	0.14	24.03	230.11	0.98	-	-	-
0–60	-	-	-		-	11.21	1.21	10
**Soil Chemical Changes**								
Before sowing	6.8	0.14	142	230.17	1.03	6.78	1.25	9.39
After harvesting	6.9	0.13	254	232.08	0.99	9.20	2.09	7.11
Difference	−0.1	0.01	−112	−1.91	0.04	−2.42	−0.84	−1.58
**Biochar**								
-	pH	Ash content (%)	Moisture wt. (%)	Volatile wt. (%)	Residual mass (char formed) wt. (%)	Total Mg (g/kg)	Organic C (%)	
	9.1	32.21	2.52	56.73	40.75	10.50	62.33	

P_2_O_5_—phosphorus pentoxide; NH_4_-N—ammonium nitrogen; NO_3_-N+N_2_O-N—nitrate plus nitrite.

**Table 2 plants-12-01002-t002:** Cumulative CO_2_–C, N_2_O–N, and CH_4_ (mg ha^−1^ hr^−1^) emissions under different treatments over the two-year study.

Treatment	Cumulative CO_2_	Cumulative N_2_O	Cumulative CH_4_
2020			
Control	13262 ± 81.71a	10.09 ± 0.17a	20.18 ± 0.20ab
N_1_	12393 ± 79.03ab	9.66 ± 0.19ab	19.78 ± 0.17ab
N_2_	12487 ± 110.1ab	9.66 ± 0.15ab	21.13 ± 0.21a
B	13685 ± 99.21a	9.66 ± 0.17ab	20.68 ± 0.13ab
N_1_B	13972 ± 83.52a	9.93 ± 0.19ab	20.77 ± 0.18ab
N_2_B	12385 ± 91.33ab	10.04 ± 0.15a	21.20 ± 0.19a
2021			
Control	8374 ± 91.01a	7.05 ± 0.11a	14.26 ± 0.20a
N_1_	8093 ± 91.26a	7.96 ± 0.11a	14.18 ± 0.18a
N_2_	8264 ± 83.21a	7.84 ± 0.13a	14.08 ± 0.20a
B	6907 ± 74.32b	6.64 ± 0.12b	13.06 ± 0.16b
N_1_B	6716 ± 63.41b	7.00 ± 0.14a	13.05 ± 0.15b
N_2_B	6833 ± 78.51b	6.97 ± 0.12a	14.16 ± 0.19a

Treatments: control-without amendments; N_1_-160 kg ha^−1^ nitrogen fertilizer; N_2_-120 kg ha^−1^ nitrogen fertilizer; B-25 t ha^−1^ biochar; N_1_B-160 kg ha^−1^ nitrogen fertilizer plus 25 t ha^−1^ biochar; N_2_B-120 kg ha^−1^ nitrogen fertilizer plus 25 t ha^−1^ biochar. Letters (a, b, and ab) show significant differences among treatments for spring barley (2020) and pea crops (2021) at *p* ≤ 0.05 (LSD).

**Table 3 plants-12-01002-t003:** Effect of biochar on global warming potential (GWP) (mg ha hr^−1^) during the years of 2020 and 2021.

Treatment	GWP of CO_2_	GWP of CH_4_	GWP of N_2_O	Cumulative GWP
2020				
Control	60.59 ± 0.117a	2.95 ± 0.05a	15.85 ± 0.15a	79.39 ± 0.112a
N_1_	61.68 ± 0.132a	2.96 ± 0.06a	15.56 ± 0.19a	80.20 ± 0.132a
N_2_	60.38 ± 0.126a	2.94 ± 0.04a	15.44 ± 0.12ab	78.76 ± 0.144ab
B	62.09 ± 0.137a	2.94 ± 0.05a	15.79 ± 0.20a	80.82 ± 0.141a
N_1_B	62.66 ± 0.118a	2.95 ± 0.02a	15.67 ± 0.14 a	81.28 ± 0.152a
N_2_B	59.58 ± 0.127a	2.96 ± 0.07a	15.73 ± 0.17a	78.27 ± 0.143ab
2021				
Control	60.81 ± 0.281a	2.82 ± 0.09a	16.29 ± 0.27ab	79.92 ± 0.138a
N_1_	59.76 ± 0.256ab	2.79 ± 0.08a	16.60 ± 0.14a	79.15 ± 0.163a
N_2_	60.77 ± 0.242ab	2.81 ± 0.07a	17.46 ± 0.21a	81.04 ± 0.140a
B	37.17 ± 0.271b	2.70 ± 0.09b	13.08 ± 0.17b	52.95 ± 0.145b
N_1_B	39.33 ± 0.229b	2.74 ± 0.02ab	15.29 ± 0.19a	57.36 ± 0.134b
N_2_B	37.70 ± 0.217b	2.75 ± 0.10ab	15.11 ± 0.21a	55.53 ± 0.122b

Treatments: control-without amendments; N_1_-160 kg ha^−1^ nitrogen fertilizer; N_2_-120 kg ha^−1^ nitrogen fertilizer; B-25 t ha^−1^ biochar; N_1_B-160 kg ha^−1^ nitrogen fertilizer plus 25 t ha^−1^ biochar; N_2_B-120 kg ha^−1^ nitrogen fertilizer plus 25 t ha^−1^ biochar. Letters (a, b, and ab) show significant differences among treatments for spring barley (2020) and pea crops (2021) at *p* ≤ 0.05 (LSD).

## Data Availability

The data that support the findings of this study are available from the corresponding author, Muhammad Ayaz, upon reasonable request.

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
