# Peer review of "Biochar with Inorganic Nitrogen Fertilizer Reduces Direct Greenhouse Gas Emission Flux from Soil"

_plants, 2023, doi:10.3390/plants12051002_

Round 1

Reviewer 1 Report

There is a two-year study on the influence of biochar amendment on GHG emission during the growth of barley and pea. The obtained results confirm the already observed trend of the influence of biochar as a soil additive. But the work lacks soil analysis after the second year, like heavy metal content, basic soil parameters, and toxic organic fraction, as well as the yield of crops, so we can see if this will have a positive long-term impact except GHG emission. I would suggest the authors provide at least the content of heavy metals and the elemental composition of the used biochar, as well as the ratio of the yield of the examined crops after the second year of fertilization compared to the first.
I also noticed some typos, so please correct the text.

Author Response

Dear Reviewer

Thank you so much for your critical and very much important review.

Just a very kind response to your comments; I have published a detailed research article on heavy metal and crop yield as part of this experiment.  e.g. (Ayaz, Muhammad, Urte Stulpinaite, Dalia Feiziene, Vita Tilvikiene, Kashif Akthar, Edita Baltrėnaitė‐Gedienė, Nerijus Striugas et al. "Pig manure digestate‐derived biochar for soil management and crop cultivation in heavy metals contaminated soil." Soil Use and Management 38, no. 2 (2022): 1307-1321).

Moreover, I have added soil chemical changes during two year. 

Additionally, we tried to focus on the issue of GHG emissions in this article only. Since this work is the part of my PhD. studies so this is still continue hopefully for couple of more years. 

The rest I have made substantial changes the article (highlighted).

Reviewer 2 Report

Dear authors, 

Theme of the paper and work is presented well but need to be revised in some aspect, kindly see my suggested in attachment to improve the paper. 

Looking forward for the revised file 

Author Response

Dear Reviewer 

We are very thankful for your intellectual review. 

We have happily considered all the comments (highlighted) and made substantial changes to the manuscript.

Round 2

Reviewer 1 Report

Corrections are satisfactory. The paper meets the quality for publication.

Author Response

On behalf of all authors and our department and LAMMC research center, we highly appreciate and thanking you for your intellectual and critical review which certainly helped us a lot  to improve our manuscript.  

Many Thanks

Reviewer 2 Report

Thanks for fruitful revision 

Author Response

Reviewer's comment: page 3 line 98: add provider of the meteorological station
Author's response: Done, highlighted in the manuscript.
Reviewer's comment: line 95: what was pea cultivar?
Author's response: "Respect cultivar" used. also mentioned in the manuscript.
Reviewer's comment: line 128: correct: Gas sampling and flux calculation
Author's response: Done
Reviewer's comment: 22 line : 2020 and 2021 treatments of fertilizers where the same? what dates were fertilized?
Author's response: Yes, treatments of the fertilizer were the same during 2020 and 2021. Dates of fertilization is mentioned in the manuscript (Highlighted). It was around 7 days prior to the sowing.
Reviewer's comment: line 153-155 : the same information is in line 97-99
Author's response: Deleted the repetition. 
Reviewer's comment: in methods no information about growth stages. it should be added in methods.
Author's response: Added
Reviewer's comment: it would be interesting to know how biochar affected barley in the second growth year, same with pea. Have you considered growing pea next year barley and measuring? or growing 2 years of barley or 2 years  pea?
Author's response: Dear sir/Madam I am extremally apologizing for being unable to add the crops yield data at this stage, as our experiment is still in progress and will continue till Sep, 2024. Actually, being a PhD. student we follow department plans for choosing crop. So, I sow spring barley in 2020 as one field trial, followed by pea crop in one field and spring barley in another field as two field trials in 2021, followed by winter wheat and pea crop in 2022 . In 2023 we will follow department plans and will continue the experiment. So our plan is to publish 4-5 year crop yield data which could be one more addition to my PhD thesis.